



# Coexistence of two dune scales in a lowland river

Judith Y. Zomer[1], Bart Vermeulen[1], and Antonius J. F. Hoitink[1]

[1]Department of Environmental Sciences, Hydrology and Quantitative Water Management Group, Wageningen University & Research, Wageningen, Netherlands

**Correspondence:** Judith Zomer (judith.zomer@wur.nl)

**Abstract.** A secondary scale of bedforms, superimposed on larger, primary dunes, has been observed in fluvial systems worldwide. This notwithstanding, very little is known about the morphological behaviour and characteristics of this secondary scale. This study aims to better characterize and understand how two dune scales coexist in fluvial systems, and how both scales adapt over time and space, considering their interdependence. The study is based on analysis of a biweekly multibeam echosounding dataset that spans a period of approximately three years and 34 kilometers in the Waal River, a lowland sand-bedded river. Results reveal that the secondary dune scale is ubiquitous across space and time, and not limited to specific flow or transport conditions. Whereas primary dunes lengthen during low flows, secondary dune height, lee slope angle and length correlate with discharge. Secondary dune size and migration strongly depend on the primary dune lee slope angle and height. Secondary dunes can migrate over the lee slope of low-angled primary dunes, and their height is inversely correlated to the upstream primary dune height and lee slope angle. In the Waal River, a lateral variation in bed grain size, attributed to shipping, largely affects dune morphology. Primary dunes are lower and less often present in the southern lane, where grain sizes are smaller. Here, secondary bedforms are more developed. At peak discharge, secondary bedforms even become the dominant scale, whereas primary dunes entirely disappear but reestablish during lower flows.

## 1 Introduction

Dunes are the key components of sand-bedded rivers and have been extensively researched to better understand and manage fluvial systems (Wilbers, 2004; Best, 2005; Parsons et al., 2005; Nittrouer et al., 2008; Cisneros et al., 2020; Naqshband and Hoitink, 2020; De Lange et al., 2021; Zomer et al., 2021). Dunes control local flow dynamics. They often cause permanent or intermittent flow separation downstream of the lee slope (Wilbers, 2004; Lefebvre et al., 2014; Best and Kostaschuk, 2002), exerting a strong control over energy dissipation. On a larger scale, this relates to hydraulic roughness (Maddux et al., 2003b, a), and is therefore vital for flood risk assessment. Turbulent flow structures generated over dunes affect sediment transport (McLean et al., 1999; Bradley et al., 2013) and dune migration is closely linked to bedload transport. Bed shear stress over the dune stoss increases towards the crest, eroding the stoss and transporting bed sediments further downstream. On the lee slope, sediments are deposited as a result of a decrease in bed shear stress (Ashley, 1990; Simons et al., 1965). Dunes are further relevant in relation to navigability, safety of infrastructure, ecology and the reconstruction of past river flow and sediment transport conditions.



Over the past decades, dune research has primarily been focused on the occurrence of one dune scale. This notwithstanding, smaller, secondary scales of bedforms have very often been observed to be superimposed on the larger dunes. Cisneros et al. (2020) have analysed bathymetric data from the Amazon, Mekong, Mississippi, Missouri, Parana, and Waal rivers and reported that superimposed bedforms are present in all those systems. In the Parana river, superimposed bedforms have been observed on most stoss slopes of larger dunes during a single field campaign by Parsons et al. (2005), and in the Mississippi river by Harbor (1998). Wilbers and Ten Brinke (2003) and Wren et al. (2023) observed superimposed bedforms emerging during the falling limb of a flood peak in the Rhine river and Mississippi river, respectively. Galeazzi et al. (2018) studied the coexistence of two dune scales in the Amazon river based on single-day field surveys at two locations during flood stage and observed that 90% of the primary dunes at water depths larger than 20 m where covered with superimposed bedforms on the stoss. The majority of dunes also featured secondary bedforms on their lee slope. Superimposed ripples and dunes have also been observed during field surveys in the Rhine river (Carling et al., 2000), the Ohio river (Baranya et al., 2023) and in the lower Yellow river (Zhang et al., 2022). Similarly, superimposed bedforms have been reported in experimental settings (Reesink et al., 2018; Venditti et al., 2005; Martin and Jerolmack, 2013).

Superimposed ripples or dunes have been considered to be a second order descriptor of the primary dunes, which are then referred to as compound dunes in stead of simple dunes (Ashley, 1990). Two different processes have been observed and described that lead to the superimposition of secondary bedforms. First, secondary bedforms have been observed to emerge during the falling limb of a flood wave (Wilbers and Ten Brinke, 2003; Wren et al., 2023). The newly emerged secondary scale is then considered to be the active bedform scale, which is in equilibrium with the decreased discharge (Martin and Jerolmack, 2013). At that stage of the discharge wave, the primary dunes become inactive and are slowly cannibalized. Secondary bedforms travel both over the stoss and lee of the primary dunes. Co-occurrence of two scales of bedforms has also been observed under steady flow conditions (Zomer et al., 2021). In steady flow, it is expected that the secondary scale develops in the boundary layer that establishes over the primary dune (Ashley, 1990). Superimposed bedforms are then often observed only on the stoss (Parsons et al., 2005). However, secondary bedforms are not limited to the primary dune stoss, but can persist over the full length of the primary dune (Galeazzi et al., 2018; Zomer et al., 2021). Secondary bedforms can have steep lee side angles (Zomer et al., 2021) and are thus likely to cause flow separation, and affect local flow and sediment transport dynamics, and also affect primary dune development (Reesink and Bridge, 2007). Both field and laboratory studies have indicated that secondary bedforms migrate comparatively fast and the bedload sediment transport associated with the small scale equals or even exceeds transport associated with primary dune migration (Zomer et al., 2021; Venditti et al., 2005).

Although superimposed bedforms have been observed in many systems, existing studies often focus on the primary scale, or are limited to flume experiments and small-scale field studies. Knowledge of secondary bedforms is still extremely limited, even though they can play a significant role in various processes, such as flow separation and roughness generation, local sediment transport dynamics and they can potentially affect the primary dune development and migration. A full characterization of superimposed bedforms is missing, leaving many, even basic, questions unanswered. A first question is whether the development of a secondary scale of bedforms is limited to specific conditions and locations, or that they are more prevalent. This determines also what the potential impact is of the secondary scale. Secondly, very little is known about the morphodynamics



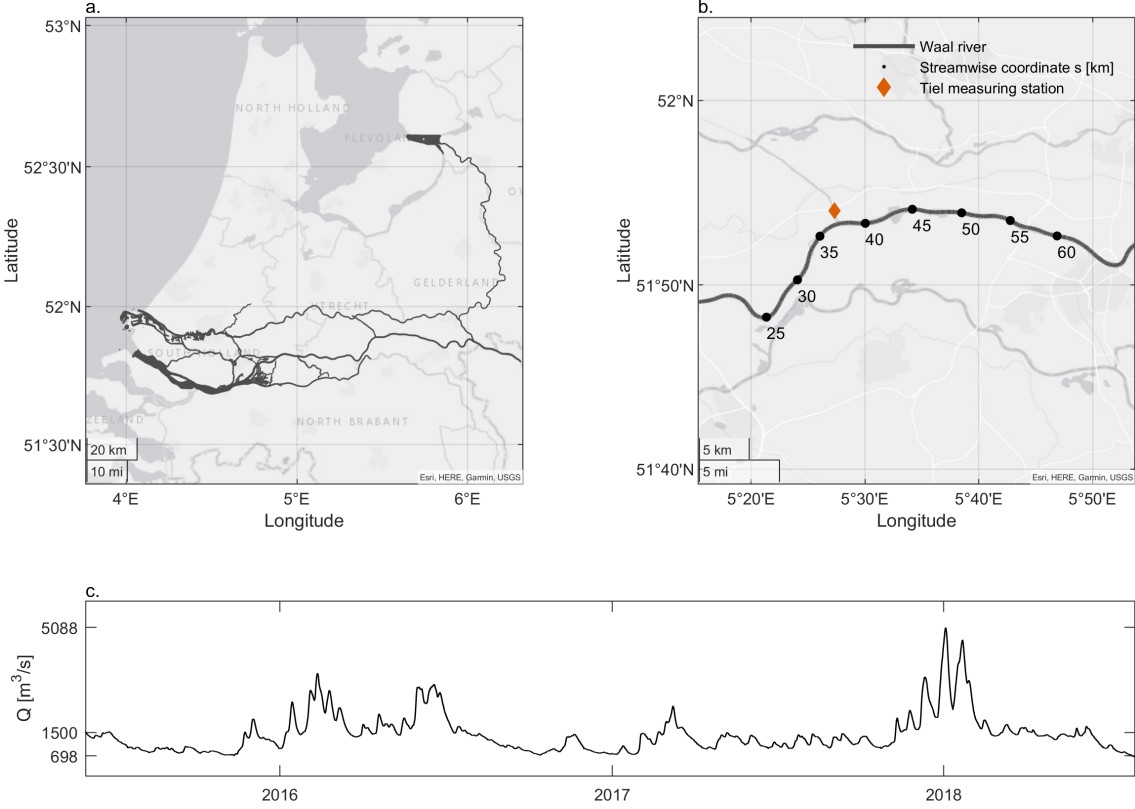

**Figure 1.** Field study area. (a) The Waal river is the main distributary of the river Rhine in the Netherlands. (b). Part of the Waal river. The study area includes kilometers 26 to 60. (c) The discharge during the time period studied, at the Tiel measuring station.

of secondary bedforms, how their properties depend on flow and transport conditions, and how they relate to primary dune morphology. Secondary dune height and lee slope angle, for example, determine the potential impact on hydraulic roughness and local sediment transport dynamics. We question how often, and under which conditions, secondary dunes migrate over the full length of the primary dunes. This has only been determined based on small-scale field studies. The persistence of

secondary bedforms impacts primary dune migration, dynamics of bedload transport, transport quantification based on dune tracking, and the interpretation of fluvial rock records. We aim to quantify how often secondary dunes persist and under which conditions. These questions are explored in the Waal River, a sand-bedded, lowland river. Bed elevation data from the Waal River are processed with a new method to separate two bedform scales and subsequently identify and characterize both primary and secondary dunes (Zomer et al., 2022).





## 2 Methods

### 2.1 Study area and description of data

The Waal River is the main branch of the River Rhine in the Netherlands (Figure 1), which carries a 30-year median discharge of approximately $1500 \, \text{m}^3 \, \text{s}^{-1}$ (Lokin et al., 2022). The Waal River connects the port of Rotterdam to the inland and Germany, and is subject to intensive shipping. The river reach subject to this study covers a distance of 38 km and a time period of 3 years (19 August 2015 to 13 July 2018). Primary and secondary dunes coexist in the Waal. Typical primary dune heights and lengths range from 0.8 to 1.5 m, and 40 to 120 m respectively. Secondary dune heights and lengths typically range from 0.10 to 0.35 m and 7 to 15 m. The bed grains significantly vary in size from north to south, offering unique conditions to study the effect of grain size on dune morphology.

Bed elevation data have been collected through Multibeam Echosounding in the fairway. The river bed has been scanned approximately fortnightly. The bed elevation data were delivered as x,y,z - point clouds with a spatial resolution of $1 \times 1 \times 1$ cm. The data were converted from a Cartesian to a curvilinear $(s, n)$ coordinate system. The $s$ coordinate is parallel to the central river axis and positive in the upstream direction. The $n$ coordinate is perpendicular to the river axis and defined as $n = 0$ at the central axis. The $n$ coordinate is defined positive towards the north river bank, and negative towards the south river bank. The bed elevation profiles located between $n = $ -70 m and $n = 70$ m are included in further processing.

The bed elevation data have been interpolated on a grid with a streamwise resolution of 0.1 m, a transverse resolution of 1 m, and a vertical resolution of 0.01 m. Interpolation was performed through inverse distance weighted interpolation, using the Geospatial Data Abstraction Library (GDAL, GDAL/OGR contributors, 2020).

Information on the bed grain size distribution has been collected in 2020 by Rijkswaterstaat. Bed samples have been acquired over the central river axis every half kilometer and north and south of the central axis every kilometer. Samples have been taken with a Hamon grab (Eleftheriou and Moore, 2013) and include material from approximately 25 cm of the top layer.

Discharge data were provided by Rijkswaterstaat for the measuring station at Tiel (Figure 1). The discharge values are calculated with a rating curve based on water level measurements.

### 2.2 Flow and transport conditions

Flow and sediment transport conditions were calculated to aid interpretation of observed dune morphology. These conditions were determined for peak, median, and low discharge levels and for the north and south of the river separately. The latter distinction was made because the bed sediments are finer in the south of the river compared to the north. Values for the discharge $(Q)$, mean flow velocity $(U)$, and water depth $(d)$ were adopted from Lokin et al. (2022). They studied the primary dunes in the Waal river, over a reach of 16.5 km, between 2011 and 2020 and determined typical flow characteristics for the study area (Lokin et al., 2022, Table 1). Since this 16.5 km reach is located in the area studied here, these flow characteristics were used to calculate sediment transport and dune characteristics. The $D_{50}$ and $D_{90}$ are median values of all individual samples in the south section of the study area $(-70 < n < -35)$ and in the north section $(35 < n < 70)$. Predicted dune height and length were also calculated to enable comparison between dune theory and observed dune morphology.



The Van Rijn transport stage parameter (Van Rijn, 1984a) was calculated as follows:

$$T_{VR} = \frac{(u'_*)^2 - (u'_{*,cr})^2}{(u'_{*,cr})^2}, \tag{1}$$

with

$$u'_* = (g^{0.5}/C')\overline{u}, \tag{2}$$

and

$$C' = 18\log(12R_b/3\mathrm{D}_{90}), \tag{3}$$

where we assume $R_b = d$, because the width of the river is much larger then the depth. $u'_{*,cr}$ is the critical shear velocity obtained from the Shields diagram (Van Rijn, 1984a), where

$$\theta_{cr} = \begin{cases} 0.04\mathrm{D}_*^{-0.1} & \text{for } 10 > \mathrm{D}_* \leq 20 \\ 0.013\mathrm{D}_*^{0.29} & \text{for } 20 > \mathrm{D}_* \leq 150, \end{cases} \tag{4}$$

with

$$\mathrm{D}_* = \mathrm{D}_{50}\left[\frac{(s-1)g}{\nu^2}\right]^{1/3}, \tag{5}$$

and

$$u'_{*,cr} = \sqrt{\theta_{cr}\mathrm{D}_{50}g(s-1)}, \tag{6}$$

Herein, $g$ is the gravitational acceleration ($\mathrm{m\,s^{-2}}$) and $s$ is the relative density. According to Van Rijn (1984c), dunes develop for $T_{VR} < 15$. For $15 < T_{VR} < 25$ there is a transitional flow regime where dunes start to wash out. For $T_{VR} > 25$, a flat bed is expected.

An alternative transport stage, T, is calculated as follows (Bradley and Venditti, 2017; Church, 2006):

$$T = \theta/\theta_{cr} \tag{7}$$

Church (2006) indicates that for $1 < T < 3.3$ bedload dominated conditions occur. For $3.3 < T < 33$ mixed-load dominated conditions, and for $T > 33$, suspended-load dominates conditions (Church, 2006; Bradley and Venditti, 2017; Venditti et al., 2016).

Another indication of when sediment suspension is expected to occur is the suspension number $u_*/w_s$. For $u_*/w_s < 1$, bedload transport is dominant, but initiation of suspension is indicated to start already at $u_*/w_s < 0.4$ (Van Rijn, 1984b). The particle fall velocity, $w_s$, is calculated following the approach of Cheng (2009):

$$w_s = \sqrt{\frac{4}{3}(s-1)\frac{g\mathrm{D}_{50}}{C_D}}, \tag{8}$$



with the drag coefficient $C_D$ defined as:

$$C_D = \frac{432}{D_*^3}(1 + 0.022 D_*^3)^{0.54} + 0.47(1 - e^{-0.15 D_*^{0.45}}). \tag{9}$$

Finally, expected dune characteristics based on the flow and sediment properties, are calculated following Van Rijn (1984c):

$$\frac{H}{d} = 0.11 \left(\frac{D_{50}}{d}\right)^{0.3} (1 - e^{-0.5 T_{VR}})(25 - T_{VR}), \tag{10}$$

and

$$L = 7.3d. \tag{11}$$

## 2.3 Separation of bedform scales and characterization of primary and secondary dunes

The bed elevation data are processed as longitudinal bed elevation profiles (BEPs). Each BEP is separated into a signal representing the superimposed, secondary dunes and the underlying bathymetry, including primary dunes, following the procedure of Zomer et al. (2022). The first step in this method is to decompose the signal using a LOESS (locally estimated scatter plot smoothing) algorithm (Schlax and Chelton, 1992; Greenslade et al., 1997). A LOESS curve is fitted to the data to separate the superimposed bedforms from the primary dunes. To avoid smoothing of steep primary lee slopes, which is unavoidable

using a continuously differentiable function, breaks are implemented at steep primary lee slopes. This is based on whether the slope of the LOESS curve at the lee exceeds a user-defined threshold. The primary lee slope is approximated with a sigmoid function. The user-defined threshold has a default value of $0.03 \, \mathrm{m \, m^{-1}}$, and is adapted per kilometer in this study based on visual inspection of the resulting bathymetric maps.

Both primary and secondary dunes are identified based on the decomposed bed elevation profiles. Dune identification is

done through zero-crossing (Van der Mark and Blom, 2007; Zomer et al., 2022). Crests and troughs are determined as local maxima and minima between two zero-crossings. To identify primary dunes, zero-crossing is done with a moving average of 4 times the estimated primary dune length.

Dune height (H) is determined as the vertical distance between the crest elevation and the average of the up- and downstream trough elevations. The dune length (L) is defined as the horizontal distance between the up- and downstream trough. The lee

slope angle ($\alpha$) is calculated as the maximum value between dune crest and trough based on a moving average of the slope. For the primary dunes, the window length of the moving average is 3 m. For secondary dunes, the window length is 0.3 m. Identified secondary and primary dunes can include fluctuations around the zero-line, which are not considered to be dunes. Primary dunes are filtered out based on the following values: $0.25 > \mathrm{H}_P > 4.0$; $25 > \mathrm{L}_P > 350$; $0.003 > \frac{\mathrm{H}_S}{\mathrm{L}_S} > 0.2$; $\alpha_P < 0.03 \, \mathrm{m \, m^{-1}}$. Secondary dunes are filtered if: $0.05 > \mathrm{H}_S > 0.75$; $0.5 > \mathrm{L}_S > 25$; $0.005 > \frac{\mathrm{H}_S}{\mathrm{L}_S} > 0.2$; $\alpha_S < 0.03 \, \mathrm{m \, m^{-1}}$; the crest elevation of a secondary

dunes is in the original signal less than 0.01 m higher than the up- or downstream trough.

Dune properties are statistically represented by the median and the first and third quartile ($Q_1$ and $Q_3$), which together divide the individual dune characteristics in four equal parts. The interquartile range (IQR) represents the spread around the median and is defined as: $\mathrm{IQR} = Q_3 - Q_1$.



The fraction of the river bed covered by dunes is computed based on the locations of the up- and downstream troughs. The
distance between the troughs is considered as covered river bed and the rest is uncovered. Based on the cover fraction, it was
determined whether a primary dune has superimposed secondary bedforms. To prevent individual or misidentified secondary
dunes from influencing the classification, the cover maps were processed towards areas with large spatial coherence. This was
done as follows. First, a two-dimensional moving average filter was applied with a window of $5 \times 5$ grid cells. All grid cells
with a value exceeding 0.5 were classified as 1 (dune cover) and with a value below 0.5 as 0 (no dune cover). Subsequently,
areas, where all neighboring cells with the same value form an area, with a size smaller than 500 cells were removed. Then, if
the primary dune lee slope was (partly) covered, it was indicated to have superimposed bedforms on the lee slope.

## 3 Results

### 3.1 Flow and sediment transport conditions

The flow and sediment transport conditions in the north side of the river, where grain sizes are larger, and in the south side
of the river, where the bed material is finer, are summarized in Table 1, including predicted dune height and length. Table 1
shows the large difference in the median grain size between the north and south river sections. This variation in grain size
leads to differences in predicted sediment transport conditions. Along the southern margin of the river, during median to peak
discharge, the transport stage parameter indicates that mixed-load transport conditions occur. The suspension number indicates
that initiation of suspension is reached. In the north section on the other hand, mixed-load conditions are not predicted, even
during peak flow. The predicted dune height varies between 0 and $1.14\,\mathrm{m}$. During low discharge, sediment may be immobile
along the northern margin of the river, and therefore, no dune development is predicted. During peak flow, the transitional
regime may be reached ($15 < T_{VR} < 25$) and dune start to wash out. Predicted dune lengths indicate that the largest wavelengths
occur during peak flow and decrease towards low flows.

**Table 1.** Flow and sediment transport conditions, predicted dune height and length in the study area for peak, median, and low discharge
levels. For each hydrodynamic condition, the first row corresponds to $D_{50}$ and $D_{90}$ on the north side and the second row to the $D_{50}$ and $D_{90}$
on south side.

|        | Q       | U       | d     | $D_{50}$ | $D_{90}$ | $D_*$ | $\theta$   | $T$        | $T_{VR}$    | $H_{VR}$  | $L_{VR}$    | $\frac{u_*}{w_s}$ |
|--------|---------|---------|-------|----------|----------|-------|------------|------------|-------------|-----------|-------------|-------------------|
|        | [m3/s]  | [m/s]   | [m]   | [mm]     | [mm]     | [-]   | [-]        | [-]        | [-]         | [m]       | [m]         | [-]               |
| Peak   | 5000    | 1-2     | >6    | 2.25     | 5.94     | 15.17 | 0.067-0.26 | 1.69-6.77  | 0.69-5.77   | 0.44-1.14 | >43.8       | 0.16-0.33         |
|        |         |         |       | 0.72     | 6.41     | 47.19 | 0.21-0.82  | 6.74-26.97 | 5.74-25.97  | 0-0.8     | >43.8       | 0.48-0.96         |
| Median | 1500    | 0.75-1.5| 4-6   | 2.25     | 5.94     | 15.17 | 0038-0.17  | 0.95-4.21  | 0-3.21      | 0-1.04    | 29.2-43.9   | 0.12-0.26         |
|        |         |         |       | 0.72     | 6.41     | 47.19 | 0.12-0.51  | 3.79-16.77 | 2.79-15.77  | 0.31-0.81 | 29.2-43.9   | 0.36-0.76         |
| Low    | 600     | 0.5-1   | 2.5-4 | 2.25     | 5.94     | 15.17 | 0.018-0.084| 0.47-2.12  | 0-1.12      | 0-0.40    | 18.25-29.2  | 0.087-0.18        |
|        |         |         |       | 0.72     | 6.41     | 47.19 | 0.057-0.26 | 1.86-8.43  | 0.86-7.43   | 0.24-0.61 | 18.25-29.2  | 0.25-0.54         |





## 3.2 Phenomenological description and occurrence of secondary bedforms in the Waal river

Figure 2 shows examples of the bed morphology where two scales of bedforms coexist. Panel a shows large, asymmetric primary dunes with lengths of around 100 m with small bedforms superimposed. In this study, we consider the superimposed bedforms as dunes and not ripples, because of their wavelength, which exceeds that of bedforms commonly described as ripples (Ashley, 1990). In most cases, the superimposed dunes visualized in panel a disappear at the primary lee slope and start to develop again at the downstream primary dune stoss. In panel b, primary dunes are shorter and higher, and secondary dunes are also larger and cover all primary dunes. In the south, superimposed bedforms continuously migrate. Panel c shows an example of a primary dune where superimposed dunes do not continuously migrate over the full dune length, but disintegrate at the lee slope. Panel d shows an example of superimposed dunes migrating over the full length of the primary dune. More generally, primary dunes in the Waal River have two- or three-dimensional crests. Crests are often tilted with respect to the channels axis, or have a barchan shape. Secondary dune crests can also be two or three dimensional but their orientation differs from the primary crest orientation. Secondary dune crests are transverse to the channel axis. Especially during lower discharge levels, there can be patches without any secondary dunes. During high discharge, secondary dunes develop over very short distances if they have disintegrated at an upstream primary lee slope and grow towards the next primary dune crest.

The fraction of the river bed covered by superimposed dunes was determined in order to assess how ubiquitous superimposed dunes are in the Waal river at two contrasting discharge levels. Figure 3 shows that at both high and low discharge levels, secondary dunes are present throughout the study area. Especially during high discharge, the majority of the river bed is covered by superimposed dunes. The cover fraction is larger towards the southern river bank during both high and low river discharge.

Figure 4, panel b, shows the cover fraction for kilometers 34.3-36 for a range of discharge levels, indicating that for the full range of discharge levels, a secondary dune scale is present and that cover correlates with discharge.

## 3.3 Temporal and spatial variability

Figure 4 further shows the primary and secondary dune characteristics for a range of discharge levels. With increasing discharge, primary height and lee slope angle increase and primary length decreases. Secondary dune length, height and lee slope angle increase with increasing discharge.

Figure 3 shows that secondary dune cover tends to be higher towards the southern river bank. This is further explored in Figure 5. For both low and high discharge, the cover fraction varies laterally. Towards the northern river bank, primary dune cover is higher than towards the south. The secondary dune cover shows an opposite pattern. Both primary and secondary dune cover are higher for higher discharge, over the full river width. Towards the northern river bank, primary dunes are higher and shorter. The lee slope angles do not vary accordingly, but are slightly larger in the northern river section. Secondary dunes, on the other hand, have slightly larger amplitudes in the southern river half at low discharge. The lateral variability in secondary dune characteristics greatly increases at high discharge. Secondary dunes are also longer and posses higher lee slope angles towards the southern river bank.



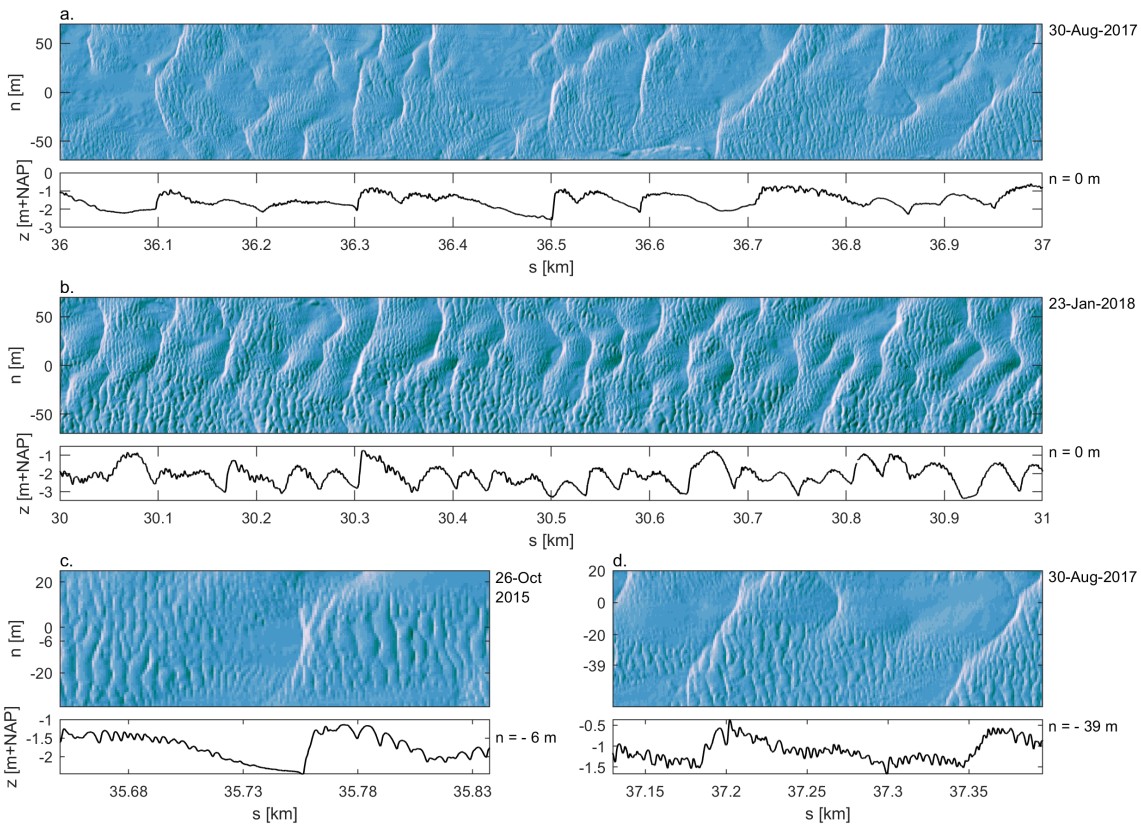

**Figure 2.** Examples of the bed morphology, indicating the coexistence of primary and secondary dunes in the Waal river. The bathymetry is visualized using hillshade. Below each bathymetric map, a BEPs from that map is shown.

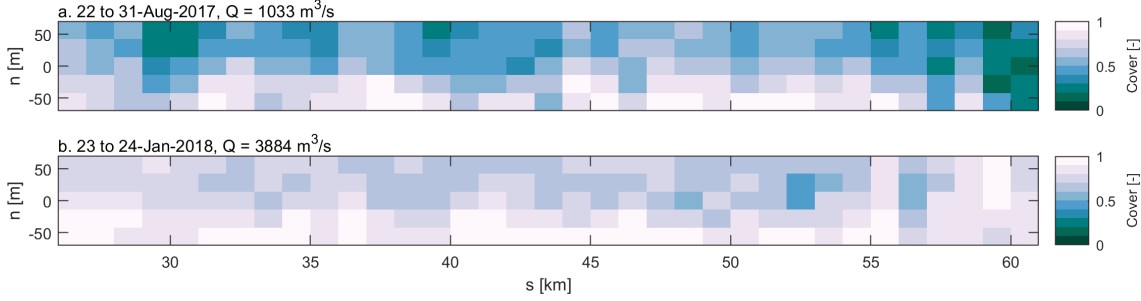

**Figure 3.** Fraction of the river bed covered by secondary dunes for two discharges. The river is laterally divided in 5 sections of equal width and longitudinally in 1 km sections. For each section, the cover fraction was determined.

Figure 6 shows that in the southern section of the river ($n = -50\,\mathrm{m}$), at peak discharge, primary dunes fully disappear and the secondary dune scale becomes dominant. At subsequent lower discharges, primary dunes reestablish. This regime shift is not observed however in the northern half of the river ($n = 50\,\mathrm{m}$). Here, during the same discharge wave, primary dunes

Earth **Surface**
**Dynamics**
Discussions



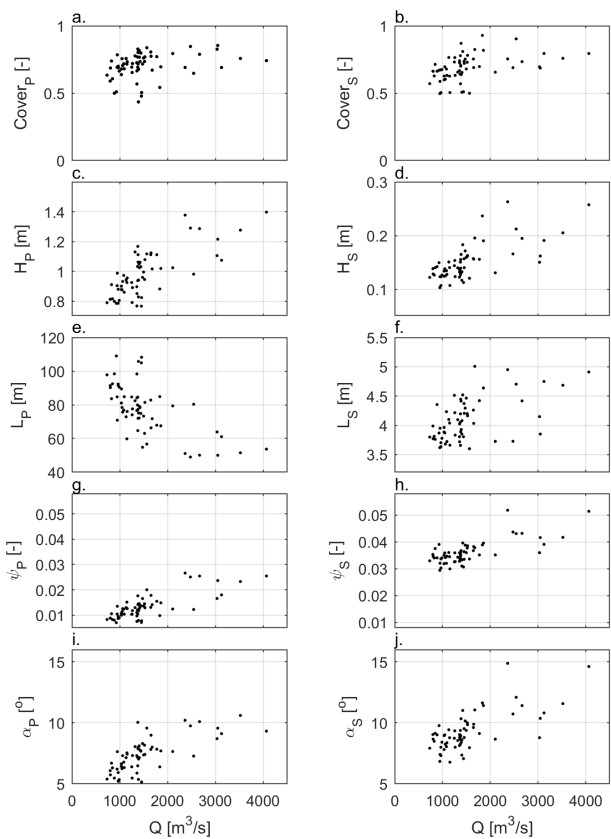

**Figure 4.** Primary and secondary dune characteristics for different discharge values. For each time step, the median of each dune characteristic was determined for the stretch between kilometers 34.3-36. The first column shows the primary dunes characteristics, the secondary column the secondary dune characteristics

become higher and shorter, with an increase in lee slope angle. Figure 7 further shows the change in dune properties over time, as determined separately for the south river section and the central and north sections. Figure 7 indicates that primary dune cover is lower in the south over the full time range, whereas the secondary dune cover is higher. During the discharge peak shown in Figure 6, along the southern margin, primary dunes do not increase in height, and dunes become longer. In the central and northern section, primary dunes become higher, shorter and lee slope angles increase. Secondary dune height in the south shows a strong peak in height, length, and lee side slope at peak discharge.

### 3.4 Secondary dune migration over the primary lee slope

Secondary dunes are observed to disintegrate at primary lee slopes in some cases, and in other cases to migrate over the full dune length (Figure 2). To further explore this, the full primary dune dataset was divided in two groups, based on the presence of secondary dunes on the lee slope. Primary dunes with less than 25% of the stoss covered with secondary dunes, were



Earth **Surface**
Dynamics
Discussions



**Figure 5.** Spatial distribution of dune characteristics. (c) Median grain size ($D_{50}$) for individual bed samples. (d-k) Lateral distribution of dune characteristics. For each BEP, the median of each characteristic was determined over the length of the study area, as well as the IQR. (i) Boxplots of the median grain size, grouped based on lateral location (north, centre, south) in the river. The boxplots indicate the median, $IQR$ and the 5th and 95th percentiles.

excluded, which was the case for 12% of the total number of primary dunes. Over 65% of primary dunes, secondary dunes do not persist. Secondary dunes migrate over the lee slopes of 24% of the total number of primary dunes. A comparison of characteristics between the two groups is shown in Figure 8. Figure 8 shows that the primary dunes over which secondary dunes not continuously migrate have higher amplitudes (panel a), slightly shorter length (panel b), and larger lee slope angles and aspect ratios (panel c). The secondary dunes that migrate over the primary dune lee slope have larger amplitudes, lengths, and lee slope angles (panels e-g). The relative dune height ($\frac{H_S}{H_P}$) is larger.







**Figure 6.** Temporal development of dune morphology over a discharge peak. (a,b) Hillshade maps at two moments. (c) Discharge. (d) Temporal evolution of BEP located at n=50 m. (d) Temporal evolution of BEP located at n=-50 m

The bottom panels (i-n) show examples of BEPs at subsequent time steps. In each of the panels, a migrating primary dune can be tracked. In panel i, the primary lee slope angle decreases and superimposed dunes start to migrate over the full length of the dune. In panel k, a reverse transition can be observed. The primary lee slope is suddenly steeper and superimposed dunes disintegrate at the lee. In panel m, both transitions can be observed.

**Figure 7.** Median dune characteristics over time. The median values are based on kilometers 34.3-36 and are determined separately for the southern section ($-70\,\mathrm{m} \leq n \leq -50\,\mathrm{m}$) and the central and northern section ($-50\,\mathrm{m} < n \leq 50\,\mathrm{m}$).

## 235 4 Discussion

### 4.1 Omnipresence of two coexisting dune scales

In literature, two distinct mechanisms are proposed that lead to the coexistence of two bedform scales. The first is that a smaller scale of dunes appears during the falling limb of a discharge peak. This smaller scale is supposed to be in equilibrium with the discharge then prevalent, and is cannibalizing the larger primary dunes, which have been formed under peak discharge 240 conditions (Martin and Jerolmack, 2013). When secondary bedforms are observed under stable discharge conditions, they





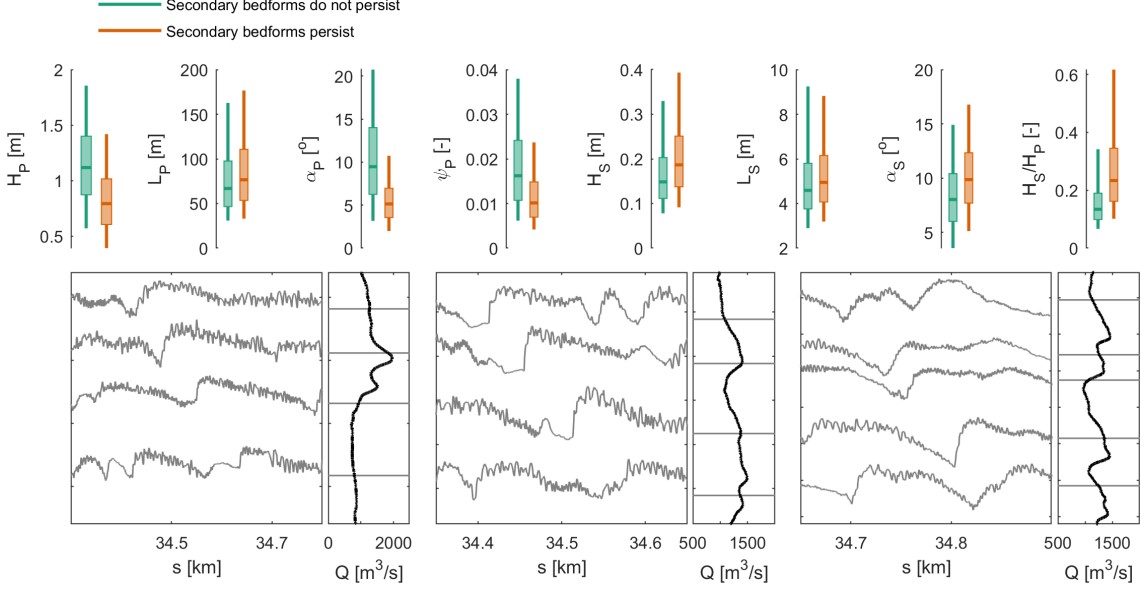

**Figure 8.** Boxplots based on the IQR and 5[th] and 90[th] percentile indicating characteristics of primary dunes with secondary dunes migrating on the lee slope (secondary bedforms persist) and primary dunes over which secondary dunes disintegrate at the lee (secondary bedforms do not persist). The median characteristics between the two groups are significantly different based on the Mann Whitney U test ($p < 0.05$). The lower plots show examples of BEPs over time where secondary dunes temporarily migrate over the primary dune lee slopes.

are assumed to develop in the boundary layer that establishes on the primary dune (Ashley, 1990). Although Wilbers and Ten Brinke (2003) previously observed superimposed dunes specifically during the falling limb of a discharge wave at an upstream location in the Dutch part of the Rhine river, in this study, secondary dunes are clearly observed over the full range of flow conditions, including peak discharge and during stable discharge conditions (Figures 3 and 4b). For the full range of

discharges included in this study, the fraction of the river bed covered by both dune scales is roughly above 0.5. The second mechanism (dunes developing in the boundary layer of primary dunes) is therefore considered to be a more likely explanation for the behavior observed in this system.

## 4.2    Primary and secondary dune morphodynamics

Mapping of the dune characteristics indicates high variability in primary and secondary dune morphology, both in time and

space. Figure 4 shows that primary dunes become higher and steeper with increasing discharge. At the highest discharge levels, this correlation is unclear. In the Waal river, at higher discharges, part of the water is discharged through the floodplains, and the shear stress does not necessarily increase with discharge (Wilbers and Ten Brinke, 2003; De Lange et al., 2021). The observation that dune height scales with discharge and thus water depth and shear stress is in line with existing literature (Yalin, 1964; Julien and Klaassen, 1995; Bradley and Venditti, 2017), although at high transport stages dunes are expected to decrease

in height, because they wash out to upper stage plane bed (Allen, 1978; Van Rijn, 1984c). Based on the estimated Van Rijn





transport stage parameter (Table 1), towards peak discharge, a transitional and even washed-out flow regime can be reached in the southern river section, where the bed sediments are finer. Indeed, Figure 5 indicates that especially for high discharge, primary heights are larger towards the northern river bank. Here, the transitional flow regime is not attained (Table 1).

Based on analysis of dune properties in a range of rivers, Cisneros et al. (2020) showed that lee-side angles increase with increasing dune height, which is also observed in this study. Cisneros et al. (2020) observed that in some rivers the largest mean lee side angles seems to decrease with increasing dune height. Van Rijn (1984c) indicates that dune steepness increases with the transport stage parameter, for values up to around $T_{VR} = 5$.

Regarding dune length, however, existing scaling laws suggest that dune length scales with water depth (Yalin, 1964; Julien and Klaassen, 1995; Van Rijn, 1984c), which is not confirmed in the area subject to study. Lokin et al. (2022) observed the same, based on analysis of primary dune properties in the Waal River. They suggest that during low flows, the migration of primary dunes becomes more diffusive, due to a larger influence of gravity in the erosion-deposition processes for smaller forcing from the flow. This has been previously observed in a modeling study by Niemann et al. (2011).

Cisneros et al. (2020) observed that dunes with steeper lee side slopes and thus permanent flow separation tend to be higher with respect to length, from which they infer that flow separation may exert a control on wavelength. This can be relevant in the Waal River, since primary lee slopes become steeper with increasing discharge, and the control on wavelength may only be present at higher discharges. Comparing the observed dune lengths to the predicted ranges of dune length according to Van Rijn (1984c), observed dune lengths are comparable to predicted dune length for peak discharge (Table 1). However, for median and low discharge, dune lengths are largely underestimated.

In the Waal River, at lower discharge levels, dunes are predominantly low-angled (Figure 4). Low-angle dunes are rarely observed in flume studies (Bradley and Venditti, 2017), though flume studies have been important in the development of existing dune theory and have contributed more to the development of scaling laws than field data (Bradley and Venditti, 2017). Considering the suggestion by Cisneros et al. (2020) that flow separation from high-angle dunes may control dune length, the mechanisms controlling the length of low-angled dunes at lower forcing levels are likely misrepresented in flume studies. It remains to be explained why the shortening of dunes with increasing discharge is observed in the Waal River and not in other fluvial systems. Secondary dune lengths do correlate with discharge.

Secondary dune height, length and lee slope angle scale with discharge. Although no scaling laws exist for superimposed dunes or ripples, it can be expected that secondary dune morphology depends on local flow and transport conditions, especially under the assumption that superimposed bedforms develop in the boundary layer that establishes on the primary dune stoss. Those local flow conditions correlate with discharge, whereas shear stress over the primary dune depends on the overall flow velocity.

### 4.3 Grain size dependence

Dune characteristics not only vary in time but are also highly variable on both short and longer spatial scales. We observe that secondary bedforms develop over very short distances and often grow towards the primary dune crest (Figure 2). This same pattern has been observed previously by Parsons et al. (2005) and Harbor (1998). The increase in size towards the primary dune





crest can relate to the increase of shear stress towards the primary dune crest (Best, 2005) and because secondary dunes have

had more space to grow when they have started to develop downstream of a primary lee slope. Carling et al. (2000) observe

in the River Rhine in Germany that superimposed dunes diminish in height at the upper primary dune stoss, likely because a

transition to upper stage plane bed (USPB) is approached at the dune crest where shear stress is highest. In this study, secondary

dune height peaks at high discharge levels and transport stage (Figure 7, panel d; Table 1). A transition to USPB is unlikely to

affect secondary dune height for the range of discharge levels observed herein, though the Van Rijn transport stage parameter

indicates that during peak discharge flat bed conditions can be reached in the southern section of the river.

    Figure 5 indicates a strong lateral variation in both primary and secondary dune cover and characteristics, especially during

high discharge. This lateral variation apparently correlates with the lateral variation in grain size of the bed material. Bed

material is fining from north to south. Wilbers (2004) has attributed the southward fining of bed material to the effect of

drawdown currents generated by passing ships. These currents are expected to cause a delivery of finer material from eroding

beaches to the channel. The drawdown currents are generally stronger in the southern river section, because ships sailing

upstream, from the port of Rotterdam to Germany, generally carry a heavier load and keep to the southern river section. An

alternative explanation could be that the frequent upstream sailing ships in the southern section temporarily reduce the bed

shear stress in the south because they oppose the downstream flow. More water is conveyed through the north section then,

increasing flow velocity and bed shear stress. Downstream sailing ships in the north section increase the bed shear stress.

    The median grain size in the north section versus the south river section significantly affects the calculated transport stage

parameter, Van Rijn transport stage parameter and the suspension number (Table 1). In the south section, initiation of suspen-

sion ($\frac{u_*}{w_s} > 0.4$) is likely during median and high flow. Similarly, the transport parameter indicates that mixed-load conditions

can be attained over the full range of discharge levels. Under such conditions, dune heights are expected to decrease, which

is in line with the observed lateral distribution. Based on the Van Rijn transport stage parameter, a transitional flow regime is

expected to be reached only during peak flow in the southern section of the river. Under those conditions, dunes start to wash

out. For peak flow, however, predicted dune heights are lower than the observed values.

    Especially the secondary dune cover and height follow a lateral pattern opposite to primary dune cover and height. Secondary

dunes are more ubiquitous, higher, longer and slightly steeper in the southern section. We expect the secondary dune size is

primarily an indirect effect of the lateral grain size variation, and it depends more directly on near-bed flow conditions that are

determined by primary dune morphology (Bennett and Best, 1995; Kwoll et al., 2016; Best and Kostaschuk, 2002; Lefebvre

et al., 2014). Downstream near-bed flow conditions depend on the primary lee slope angle (Bennett and Best, 1995; Kwoll et al.,

2016; Best and Kostaschuk, 2002). Especially downstream of steeper lee slope angles, the turbulent wake that develops might

limit the vertical space in which secondary dunes develop (Zomer et al., 2021). Also, a larger relative dune height increases

the length of the flow separation zone and turbulence production (Lefebvre et al., 2014). Flow over flatter, low-angle dunes

will more closely resemble flow over a flat bed (Kwoll et al., 2016), leading to more favorable conditions for secondary dune

development (Liu et al., 2020). Often, secondary dunes persist more over flatter dunes with low lee slope angles (Figure 8),

creating more space for secondary bedforms to develop.





A recent study in the Waal River has indicated previously that secondary dune height is inversely correlated with primary
dune height (Zomer et al., 2021). In the present study, the correlation has been determined between the upstream primary
dune height and lee slope angle on one hand, and the average secondary dune height on the downstream primary dune on the
other. This has been done for each timestep seperately. For primary dune height, the Pearson correlation coefficients that are
significant ($p < 0.05$), range between -0.042 and -0.36, with a median value of -0.19. For primary dune lee slope angle, the
Pearson correlation coefficients range between 0 and -0.43, with one outlier of 0.17. The median value is -0.20. The lowest
correlation coefficients are found at higher discharge levels.

In a more extreme case, we observe that during peak discharge, primary dunes almost fully disappear, and the superimposed
dunes become the dominant bedform scale in the southern section of the river (Figure 6). As far as we are aware, such regime
shift has not been reported before for other fluvial systems. The observation that superimposed bedforms become dominant,
can be expected in the situation where they emerge during falling discharge and cannibalize the primary dunes. However, that
is not the case here. The secondary scale becomes dominant during high discharge and during lower flows, primary dunes
reestablish. Interestingly, in the northern section of the river, this regime shift does not occur. Here, primary dunes become
higher, shorter and steeper.

Figure 6 shows an example of a river section accommodating an inner bend bar. Although both river curvature and the
presence of bars might affect dune properties (Harbor, 1998; de Ruijsscher et al., 2020), the increased dune cover towards
the southern river bank is observed throughout the entire study area (Figure 5), and not limited to inner bends. More likely,
the lateral variation in the grain size distribution exerts a dominant control. This variation relates to the decreased presence
and height of primary dunes and increased secondary dune cover and size. The lateral variation in grain size causes a lateral
variation in sediment transport conditions, with a higher level of suspended sediment transport in the south (Table 1). Figures 6
and 7 indicate that before the discharge peak where the regime shift occurs, primary dunes are flatter and have lower lee slope
angles in the south. Secondary dunes are slightly larger throughout the observed time period, but also show a stronger response
to the discharge peaks between 2016 and 2018. The dependence of secondary dune size on primary dune morphology explains
why secondary dunes are larger in the southern section. Next to this, secondary dunes may develop more easily in finer grain
beds and under mixed-load conditions. Once the secondary dunes have developed to a larger relative dune height (compared
to primary dunes) they may become dominant in controlling local flow conditions. Reesink and Bridge (2007) have indicated
that superimposed bedforms with heights that exceed 25-30% of the primary dune height, reduce the primary lee slope. These
factors may lead to a tipping point, where the secondary dunes become the dominant scale. It is debatable if a regime shift
really occurs, since primary dunes reestablish as soon as discharge decreases, and thus the observed transition to a situation
where the secondary dune scale is dominant is reversible.

## 4.4 Secondary dune migration over the primary lee slope

Figure 2 shows that, occasionally, superimposed dunes disappear over primary lee slopes, and develop further downstream. In
other cases, the superimposed dunes persist. Although superimposed bedforms are often reported to be limited to the primary
dune stoss (Parsons et al., 2005), the persistence over the primary lee has been observed previously in field studies (Galeazzi





et al., 2018; Zomer et al., 2021). The present study deomonstrates that secondary bedforms' persistence over primary lee sides can be a common phenomenon. This seems primarily linked to the primary dune lee slope angle (Figure 8) and to a lesser extent

to the primary dune height. Both Zomer et al. (2021) and Galeazzi et al. (2018) have linked the migration of superimposed bedforms over the primary dune lee slope to the lee side angle, where Galeazzi et al. (2018) found a cutoff value of 15-18° and Zomer et al. (2021) a value of 11°. In this study, the persistance of superimposed dunes over the primary lee slope is for 95% limited to slopes with a angle below 10.7°. The strong control of the primary lee slope angle is attributed to sediment avalanching over slopes near the angle of repose and to the occurrence of flow separation over high-angle dunes. Figure 8c

indicates that sometimes, secondary bedforms disintegrate over lee slope with an angle lower than 10.7°.

The lee side slope angle largely determines whether flow separation occurs, yet this is also dependent on the relative dune height (Lefebvre et al., 2016) and might also depend on the crest line shape (Best, 2005), flow velocity, and the bed grain size distribution. Furthermore, lee slope angles have been determined parallel to the rivers axis, whereas the steepest slopes can have a slightly different orientation, which leads to an underestimation in the results. Figure 8 indicates that the superimposed dunes

that persist are more developed (i.e., are higher, longer, steeper). The more developed superimposed bedforms may control whether they persist over primary lee slopes because they have more influence on local flow conditions. However, this can also be a result of spatial correlation between primary dune characteristics. Secondary bedforms that persist may have persisted over the upstream primary dune, and thus have had more time to grow towards equilibrium size.

A key question is whether primary dunes with lower lee slope angles allow superimposed bedforms to persist, or whether

superimposed bedforms that persist decrease the primary lee slope angle. Figure 8 shows that a primary dune's lee slope angle changes over time. Steeper lee slopes without superimposed dunes transition to less steep slopes with superimposed dunes and vice versa. The observations indicate that this is not a one-way process, which suggests that the primary dune morphology controls whether secondary bedforms persist.

## 5   Conclusions

In the Waal River, secondary bedforms are ubiquitous across the full range of discharge levels included in the study. This indicates that often two scales of dunes are active simultaneously, rather than that a smaller scale cannibalizes primary dunes following peak flow. The secondary scale develops in the boundary layer that establishes on the primary dune. The size and omnipresence of secondary dunes imply a large impact on local flow processes, sediment transport dynamics, river bed development and hydraulic roughness.

The characteristics of secondary dunes are highly variable over space and time. On average, secondary dune height, length and lee slope angle increase with discharge. Secondary lee slope angles develop up to around 20°, likely causing flow separation and high turbulence kinetic energy production. This can significantly add to the hydraulic roughness in the system, and may also affect the morphological development of the primary dunes on which these secondary dunes are superimposed. Primary dunes become higher and steeper with increasing discharge, but their wavelength increases at lower discharge levels.



The morphology and spatial distribution of secondary dunes is highly dependent on primary dune morphology. Secondary dune size is inversely correlated to the height of upstream primary dunes. In the Waal River, the bed sediments are finer towards the southern river bank, which is attributed to shipping. As a result, mixed bedload-suspended load sediment transport conditions occur at lower discharge levels compared to the northern river section. This lateral variation correlates with lower primary dune cover and heights in the south, where secondary dunes, on the other hand, are larger and more ubiquitous. During

peak discharge, the secondary dunes are even observed to become the dominant dune scale in the southern half of the river. Once water levels drop, primary dunes reestablish.

Secondary bedforms are observed to disintegrate at primary dune lee slopes, which is attributed to sediment avalanching or instability of the boundary layer in which the secondary dunes have developed. In some cases, secondary bedforms are observed to migrate over the full length of the primary dune. This happens when primary dune lee slope angles are low (<

10.6°). The ability of secondary bedforms to migrate over the full length of the primary dune varies in time, depending on temporal variations in the primary lee slope.

*Code and data availability.* The MATLAB code of the bedforms separation and identification method used in this study can be accessed through https://doi.org/10.5281/zenodo.6949082 or https://github.com/j-zomer/BedformSeparation_Identification (last access: 1 August 2022).

The bed elevation data (point clouds based on multibeam echosounding data) and bed grain size data can be retrieved through the service desk data of Rijkswaterstaat (https://www.rijkswaterstaat.nl/formulieren/contactformulier-servicedesk-data). The discharge data from the station Tiel can be downloaded from https://waterinfo.rws.nl/#!/kaart/Afvoer/Debiet___20Oppervlaktewater___20m3___2Fs.

*Author contributions.* Initiation and design of the study was a result of discussion between all co-authors. JYZ analyzed the data and wrote the manuscript. The manuscript was reviewed and edited by all co-authors.

*Competing interests.* There are no competing interests.

*Acknowledgements.* This study is part of the research program Rivers2Morrow, which is funded by the Dutch Ministry of Infrastructure and Water Management and its executive organization Rijkswaterstaat. Multibeam echosounding data, discharge data and bed grain size data have been made available by Rijkswaterstaat. A.J.F. Hoitink was partially funded by the Netherlands Organization for Scientific Research (NWO), within Vici project "Deltas out of shape: regime changes of sediment dynamics in tide-influenced deltas" (Grant NWO-TTW 17062).



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
