# Peer review of "Coexistence of two dune scales in a lowland river"

_Earth Surface Dynamics, 2023_

## Referee Comment (RC1)

Dear Editor,

I have completed my review of the manuscript entitled "Coexistence of two dune scales in a lowland river" and I am pleased to recommend its publication with revisions. The paper represents a significant contribution to the field of bedform morphodynamics and carries implications for various related disciplines, including geomorphology, sediment transport, hydrology, river modeling, and sedimentology.

The major contribution of this paper lies in its quantification of the spatial and temporal range of secondary and primary bedform coexistence in a natural river. Despite the inherent complexities of the River Waal, which arise from navigational requirements, shipping impacts, and variations in grain size within the channel, the study provides valuable insights into the probable bedform dynamics observed in natural systems. The ability to compare two distinct cases, characterized by different sediment types, under similar conditions is a novel and significant contribution.

I particularly appreciated the meticulous analysis of bedform morphology, coverage, and the comparisons made with river discharge and estimated sediment transport and morphology. The paper offers a substantial advancement in our understanding of the interactions and coexistence of two bedform scales in rivers. It calls attention to the secondary scale, which is often overlooked, despite its prevalence in rivers, as convincingly demonstrated by the authors.

While the manuscript is commendable, I have a few comments in two areas. Firstly, I would like to seek further clarification on the method used to calculate coverage. Although the paper mentions the calculation of coverage for both primary and secondary scales, I feel that certain essential details are missing, preventing me from fully comprehending the procedure. It would be beneficial if the authors could explain how they derived the value representing secondary dune coverage over primary dunes, as this information is crucial for proper interpretation of the results.

Secondly, I noticed a few instances in the manuscript where the discussion lacked consideration of response times for dune morphodynamics, specifically the time lag between flow changes and resulting sediment transport and morphological changes. Addressing this aspect could enhance the discussion, particularly when comparing the morphology of secondary and primary dunes during peak flow to the dune morphology presented in other papers and to the equilibrium predicted values. While I understand that delving into a detailed discussion of response times for dune morphodynamics may be beyond the scope of this paper, I believe that incorporating these ideas into the discussion (particularly lines 270-273 and 310-311) would be valuable. Even more, in section 4.1 of the discussion, the authors suggest that secondary dunes develop in the boundary layer of primary dunes rather than forming during the falling stage. In this context, I would like to inquire about the timescale at which the superimposed dunes reach a point of response (particularly decay) before another flow fluctuation occurs that forces the growth of secondary bedforms. Is there a period within this timescale where the superimposed

dunes migrate multiple wavelengths without significant flow fluctuations that would stimulate their growth or decay, and thus indicates their development dependence within the boundary layer?

In conclusion, I recommend accepting the paper for publication with revisions. The manuscript makes a significant contribution to our understanding of bedform morphodynamics in natural river systems and has been well-written, featuring clear takeaways and beautiful figures. I commend the authors for tackling such a complex dataset and presenting their findings in an exemplary manner. It was a pleasure to read their work.

Please find my line-by-line comments below.

Sincerely,

Julia Cisneros

Jackson School of Geosciences, UT Austin

Line by line comments:

40          in stead -> "instead"

83-84          It is unclear how many profiles were analyzed. Can you be more specific here? Were they taken at a particular spacing?

106          I may be mistaken but I have checked throughout the manuscript and do not see what $\bar{u}$ is defined as. Since you define every other variable, it will be helpful to have this defined as well.

120          See the above comment for $\theta$

146-147          Why is the moving average 4 times the estimated primary length? Is there a reason or is this just a choice?

151          I'm confused why the window length for primary and secondary dunes is different. Does this affect the resultant value for leeside angle of the primary and secondary? It seems to me that the primary dunes leeside angle would be lowered by having a larger smoothing window?

153-154          Do all of the conditions have to be met in order to not be filtered (e.g. 0.25 > HP > 4.0 AND 25 > LP > 350 AND 0.003 > HS > 0.2 AND $\alpha$P < 0.03 m m$-1$) or not?

155        I'm not sure I understand this: "the crest elevation of a secondary dunes is in the original signal less than 0.01 m higher than the up- or downstream trough". Can you re-write this to be more clear?

159-166        This paragraph is hard for me to understand. Was the fraction computed for both the primary and secondary dunes? You say "based on the cover fraction, it was determined whether a primary dune has superimposed secondary bedforms." How was this done? Then the description of a 2D moving average is presented, so this is a secondary smoothing and then it looks like a third (?) kind of averaging (removing areas less than 500 cell) to get to the final grid presented. I think, if presented more clearly this is a clever way to get towards superimposed dune coverage! So, I think it is important here to make sure this method is very clear so the reader can make sense of the data presented.

188-189        I wonder, do the groynes along the river banks play a role in influencing the barchan shapes of the primary crests? I remember in this river, the groynes are oriented similarly to the primary dunes crests and have large, flame structured erosional areas that extend from downstream the groyne and into the channel fairway.

191-192        You say the secondary dunes "grow towards the next primary dune." Do you have any ideas as to how the superimposed bedforms grow along the stoss? Do they grow or decay in size as they move along the stoss of the primary dune?

Figure 2        Can you add a line on the maps to show where the profile is taken from? I know you include the "n" location, but I think adding a line may help to visually connect the subplots.

Figure 4        Can you put the variable name beside the variable symbol here (e.g. Primary dune height, $H_p$)?

Figure 5        It is unclear what the different colors mean here as there is no label and no description of these colors in the figure caption. I also believe the subplot labels are wrongly referred to in the figure caption (e.g. (c) Median grain size should be (a) Median grainsize). There is also no mention of subplots (b) and (c) in the caption.

Figure 6        There are no figure subplot labels on the figures but they are mentioned in the figure caption. Please be sure to check that subplot labels and reference to them exist and are correct in all figures.

245-247        See comment above in my general thoughts about possibly discussing the response time of superimposed dunes to the fluctuations in flow and sediment transport to be sure of this statement.

301-305        This is an interesting line of thought. Do you have any ideas about the influence of the ships traveling heavier and thus lower in the water column on the south side (lower

under keel clearance). Wouldn't this possibly increase shear stress in the southern side as the boats are passing? Just a thought!

351-353    I think this is very spot on! Very compelling and this really reinforces the "cat and mouse" problem between the competing processes at play here!

---

## Referee Comment (RC2)

Dear authors,

Review of the manuscript: Coexistence of two dune scales in a lowland river, by Zomer, Vermeulen and Hoitink.

I have read the paper with pleasure and recommend publication after minor revisions. This clearly written manuscript presents interesting findings about the morphology and behaviour of secondary dunes in a river stretch of the river Rhine in the Netherlands. This topic is very interesting for the audience of eSurf.

The conclusions are interesting and novel and well supported by the results. The manuscript is clearly written and edited. Figures are clear and informative, although explanation in the captions could be extended, such that readers can understand the figures without reading the text. The manuscript presents new and clear messages.

I have two small concerns that should be addressed.

1) Although the study area is 38 km and data covers 3 years (bi-weekly), several observations and associated conclusions seem to have been drawn from small sections or a few snapshots in time. For example, figure 4 (only km 34-36) or figure 5 (only high and low discharge). I understand that not all data can be shown, but did the authors check their findings for other locations/periods? This at least warrants a discussion about the validity of the findings for other locations in the study area or comparison with other high/low discharges.

2) In their method the authors briefly describe criteria for identifying dunes (L151-155). However, the exclusion criteria might have a significant impact on the results of for example the coverage, dune characteristics, etc. It effectively means that authors are defining a certain area as flat bed, which they do not show in their results. It would be valuable to know if these flat bed area's are connected or exits within a dune field. Do secondary dunes persist on these flat beds or not? This might also affect the methodology: how is a dune length determined near a spatial transition to flat bed. This at least warrants a discussion.

Minor comments:

P9, Fig.2. Please state the discharge for each panel in the caption.

P8,L190. "Secondary dune crests are transverse to the channel axis". Where can readers see this?

P8,L199. "… cover correlates with discharge". The correlation seems rather weak, to what extent is this correlation significant?

P8,L198. "… cover fraction for kilometers 34.3-36 …" Why do the authors only show results for this section, the study area was 38 km (L74). Please explain if these findings (L198-199) are also valid for other sections of the studied river stretch?

P10,Fig.4. Please explain the difference between panel a and panel b. Is panel a showing the coverage of primary dunes? If so, please explain how the primary dune cover was determined and how the authors explain that for some locations primary dune coverage was below 50%. Is this related to the selection criteria mentioned on L154-155? Which would imply that areas with dunes lower than 25 cm or shorter

than 25m are considered flat bed? It would be helpful if authors state explicitly that flat bed occurs for a significant percentage of the river stretch.

P10,L218. "… and dunes become longer." Where can readers see this? Figure 5 shows that primary dune length is comparable (slightly lower) at n=-50 for high discharges. Please explain.

P11,Fig.5. Please explain orange vs. green line in the caption. In caption panel a shows D50 not panel (c).

P12,Fig 6. Please add panel numbers c,d,e to the figure. (e) is not mentioned in the caption.

P14,L245-247. For which grain sizes is this statement valid? Is it possible that the difference with the findings of Wilbers and Ten Brinke (upstream Rhine, larger grain size) can be explained by grain size differences?

P14,L255. More recent studies exist about transitions to USPB, e.g. work of Naqshband (http://doi.wiley.com/10.1002/esp.3789) and Van Duin (https://www.mdpi.com/2076-3417/11/23/11212).

P15,L261. Is this observation of Van Rijn also observed in this study?

---

## Author Comment (AC1)

**Response to the reviews of 'Coexistence of two dune scales in a lowland river' by J.Y. Zomer et al.**

We thank the reviewers for their positive evaluation of our manuscript, and the useful suggestions for improvement. Below, we respond to the reviewer comments.

**RC1**

Many thanks for the support.

**Main comments:**

> 1). Firstly, I would like to seek further clarification on the method used to calculate coverage. Although the paper mentions the calculation of coverage for both primary and secondary scales, I feel that certain essential details are missing, preventing me from fully comprehending the procedure. It would be beneficial if the authors could explain how they derived the value representing secondary dune coverage over primary dunes, as this information is crucial for proper interpretation of the results.

In the procedure, cover maps are created based on locations of the troughs of identified dunes. In between troughs, the bed is considered to be covered (and assigned the value 1), the remaining area is assigned the value 0. Based on the cover maps, it is determined whether a primary dunes has secondary bedforms superimposed. To prevent individual or misidentified secondary dunes from influencing the classification, the cover maps were processed to include areas with a large spatial coherence. This was done according to the following procedure In the first step, for the two-dimensional moving average, we use the conv2 function (below) in Matlab, where B is the gridded bed elevation data, and A defines the (unweighted) two-dimensional window. ('A=ones(5,5)/5^2' in Matlab).

$$C(j, k) = \sum_{p} \sum_{q} A(p, q) B(j - p + 1, k - q + 1)$$

The second step is a filtering step, where 'small areas' are removed using the bwareafilt function in matlab. The bwareafilt function extracts connected components from a binary image, where the area of the objects is in the specified range. Subsequently, a distinction is made between 'simple dunes' (primary dunes with no secondary bedforms superimposed), primary dunes over which secondary bedforms persist (primary dune lee is fully covered) and primary dunes over which secondary bedforms do not persist, but disintegrate at the lee slope.

I changed the section to:

'The fraction of the river bed covered by dunes is computed based on the locations of the up- and downstream troughs. The distance between the troughs is considered as river bed covered by dunes and the rest is considered as uncovered. Based on the full grid containing all BEPs, this leads to cover maps with ones indicating that a dune is present and zeros if no dune is present at a grid cell. This was done for both secondary and primary dunes.

Based on the cover maps, it was determined whether a primary dune has superimposed secondary bedforms. To prevent individual or misidentified secondary dunes from influencing the classification, the cover maps were processed to only include areas with at least 50% coverage within a search window of 5 m^2 covering a 5x5 window of unfiltered coverage data. This was done as follows. First, a two-dimensional moving average filter was applied with a window of 5×5 grid cells. From the result, all grid cells with a value exceeding 0.5 were classified as 1 (dune cover) and with a value below 0.5 as 0 (no dune cover).

Secondly, all contiguous areas with a size smaller than 500 grid cells were removed. If less than 0.25 percent of the primary dune stoss is covered with secondary bedforms, the primary dunes is considered

a simple dune. If the primary dune lee side is fully covered by secondary dunes, secondary dunes are considered to persist over the primary lee slope. If the primary dune lee slope is not fully covered, secondary dunes do not persist.'

> 2). Secondly, I noticed a few instances in the manuscript where the discussion lacked consideration of response times for dune morphodynamics, specifically the time lag between flow changes and resulting sediment transport and morphological changes. Addressing this aspect could enhance the discussion, particularly when comparing the morphology of secondary and primary dunes during peak flow to the dune morphology presented in other papers and to the equilibrium predicted values. While I understand that delving into a detailed discussion of response times for dune morphodynamics may be beyond the scope of this paper, I believe that incorporating these ideas into the discussion (particularly lines 270-273 and 310-311) would be valuable.

We agree that the response time of dune morphodynamics is a relevant point of discussion, when comparing observed dune characteristics to predicted values. During the analysis phase of this study, we have attempted to visualize hysteresis patterns, but these have not been observed. There was no clear distinction between dune characteristics observed during the rising of falling limb of a discharge wave. Wilbers and Ten Brinke (2003) did observe hysteresis. They observed that the maximum dune height lagged behind the peak discharge with a few days. Considering that the temporal resolution of the dataset studied here is roughly fortnightly, it might be that existing hysteresis patterns during a discharge peak can simply not be determined because the temporal resolution is too low.

Zomer et al. (2021) report a field study from the river Waal where two dune scales where observed during stable, low flow conditions. They determined whether observed primary dunes were in equilibrium with the prevailing flow conditions, using the framework of Myrow et al. (2018). They showed that the primary dunes had fully adapted after around 127 days (P95). This is a long period, but the so-called bedform turnover timescale (Myrow et al., 2018) is highly dependent on the sediment transport rate, which was very low for the conditions of that field study.

We included a short section in the discussion about the response time of dune morphodynamics at line 280:

It is important to note that primary dune characteristics are likely not in equilibrium with the flow conditions during a flood wave, hindering very exact comparison with the predicted values. In this study, no clear hysteresis patterns have been observed. Wilbers and Ten Brinke (2003) however observed hysteresis in the River Waal. In their study, dune heights increased until a few days after a discharge peak. It is likely that the temporal resolution of the dataset that is studied here, is not sufficiently high to capture the hysteresis effects near a peak. The adaption time of dunes depends on the dune size and sediment transport rate (Myrow et al., 2018), where the adaption time increases with dune size and decreases with sediment transport rate. As a result, especially following a large discharge peak, it can take a long time for the large, primary dunes to fully adapt to the prevailing, low flow conditions following a peak. Zomer et al. (2021) determined this adaptation period can be up to a few months even in the River Waal. Figure 7 shows that primary dune characteristics in this study adapt to changing flow conditions quite quickly, even though full equilibrium might not be reached.

> Even more, in section 4.1 of the discussion, the authors suggest that secondary dunes develop in the boundary layer of primary dunes rather than forming during the falling stage. In this context, I would like to inquire about the timescale at which the superimposed dunes reach a point of response (particularly decay) before another flow fluctuation occurs that forces the growth of secondary bedforms. Is there a period within this timescale where the superimposed dunes migrate multiple wavelengths without significant flow fluctuations that would stimulate their growth or decay, and thus indicates their development dependence within the boundary layer?

In section 4.1 we mainly aim to distinguish between the one mechanism where secondary dunes emerge during the falling limb of a discharge peak, where only one (the smaller) dune scale is active and the larger dunes are relicts and the second mechanism where both dune scales are actively migrating and are, at some point in time, both in (semi)equilibrium with the prevailing flow conditions. We changed the last sentence of section 4.1 to 'The second mechanism (*with superimposed dunes developing throughout all phases of the discharge peak*) is therefore considered to be a more likely explanation for the behavior observed in this system.'

We further assessed the timescale at which superimposed dunes have adapted to changed flow conditions, using the framework presented by Myrow et al. (2018). We added the following after line 285:

'We expect that the secondary bedforms are mostly in (near-)equilibrium with the prevailing flow conditions, and observed small-scale spatial patterns in secondary dune characteristics relate to the local flow structure governed by the primary dunes (i.e. the boundary layer). Myrow et al. (2018) states that the timescale governing the adjustment of bedform morphology ($T_t$) to prevailing flow conditions is related to the sediment transport rate and the bedform volume: $T_t = LH\beta/q_s$, where $\beta$ is the dune shape factor, here assumed to be 0.55 (Ten Brinke et al., 1999). Based on values presented in Figure 5 and bedload transport rates in the River Waal, presented by Wilbers and Ten Brinke (2003, Figure 13, panel C), we calculated very rough estimates of the turnover timescales. For low discharge (Q=1003m/s$^3$), Tt = 4.0 × 0.2 × 0.55/1.0 × 24 = 10.6 hours and for high discharge (Q=3884m/s$^3$), Tt = 8.0 × 0.4 × 0.55/5.0 × 24 = 8.4 hours. These estimates show that the secondary dunes adapt quickly to changing flow conditions.'

**Line by line comments:**

Line 40: Corrected.

Lines 83-84: The data were interpolated on a grid with a transverse spacing of 1m (line 85-86). For more clarity, I added 'with a transverse spacing of 1 m' to line 135.

Line 106: $\overline{u}$ is the mean flow velocity. In line 97 this is indicated with $U$, so I changed $\overline{u}$ to $U$ in line 106.

Line 120: $\theta$ is the Shields number (dimensionless). I added 'where $\theta_{cr}$ is the critical Shields number (-)' to line 112 and 'where $\theta$ is the Shields number (-)' to line 121.

Lines 146-147: This is a choice. Visual inspection of the moving-average signal and subsequently identified crests and troughs indicate four times the primary dune length gives a good result.

Line 151: We chose to scale the window with the bedform size (roughly). The reason to use a larger window size for the primary dunes is that this has a stronger link to the corresponding flow patterns and based on our own analysis is also a better predictor for whether secondary bedforms persist or not, compared to using a very small window. Considering the lengths of primary dunes, the larger window should still be able to capture the slope of the slip face, the steepest section of the lee side.

Lines 153-154: No, one of the conditions has to be met. To clarify, I changed 'based on the following values' in line 153 to 'if one of the following conditions is met'.

Line 155: I changed this to: 'if for an identified bedform, the crest height in the unfiltered BEP is less than 0.01 m higher than the up- or downstream trough height. The last condition can be met for example when a small fluctuation exists at a steep primary dune lee slope.'

Lines 159-166: See the response to the first main comment.

Lines 188-189: It is an interesting comment whether the groynes affect the primary crest shape. We did not look into this. In this analysis, the study area is restricted to the thalweg/navigation channel and the areas towards the river banks where scours occur are excluded. Also, the primary dunes are not always

barchan shaped, but can be more two-dimensional (though then often slanted). I would think there is at the least an indirect effect because groynes affect the cross-sectional flow. This effect might be difficult to distinguish from effects of channel bends, bars, grain size variability and such.

Lines 191-192: Grow in size, I added this into the sentence.

Figure 2: I added lines on the maps.

Figure 4: I have written the full variable names in the caption, along with the variable symbol.

Figure 5: I added a legend to explain the colors. Indeed, the subplot labels do not match the caption. I changed the caption to:

'Spatial distribution of dune characteristics for two discharge level. (a) Boxplots of the median grain size, grouped based on lateral location (north, centre, south) in the river. The boxplots indicate the median, IQR and the $5^{th}$ and $95^{th}$ percentiles.(b-k) Lateral distribution of primary and secondary dune characteristics. Characteristics include fraction of the river bed covered by primary dunes ($cover_p$) and secondary dunes ($cover_s$), dune height (H), length (L), aspect ratio ($\psi$), and lee slope angle ($\alpha$). For each BEP, the median of each characteristic was determined over the length of the study area, as well as the IQR.'

Figure 6: I added labels to the subplots in figure 6.

Lines 245-247: See the response to the second main comment.

Lines 301-305: I know there is some ongoing research on the effects of shipping on dune characteristics in the River Waal, proving that this question has not been answered yet. As for the effect of ships on the grain size distribution (and indirectly on dune characteristics), I agree that the under keel clearance will affect the shear stress. However, a larger shear stress would result in larger grain sizes (more transport capacity), whereas the bed material is finer in the south.

Lines 351-353: Thank you for the positive feedback!

**References**

Myrow, P. M., Jerolmack, D. J., & Perron, J. T. (2018). Bedform disequilibrium. Journal of Sedimentary Research, 88(9), 1096-1113.

**RC2**

Many thanks for the support.

**Main comments:**

1) Although the study area is 38 km and data covers 3 years (bi-weekly), several observations and associated conclusions seem to have been drawn from small sections or a few snapshots in time. For example, figure 4 (only km 34-36) or figure 5 (only high and low discharge). I understand that not all data can be shown, but did the authors check their findings for other locations/periods? This at least warrants a discussion about the validity of the findings for other locations in the study area or comparison with other high/low discharges.

We did not analyze the full dataset, because the computational time is very large, especially the processing step where the x,y,z-point clouds were interpolated on a grid. We chose to analyze the full reach of 34km for two contrasting discharge levels and the full time period (3 years) for a smaller reach of 2.7 kilometers. For most findings it is likely that they are valid for the full dataset, because they are supported by both subsets of the data. For example, the conclusion that secondary bedforms are omnipresent or the primary dune lengths are larger during low discharge levels. On the other hand, the data presented in Figure 6 we consider to be site-specific and should be presented as such.

Though the results clearly indicate when the presented results are either based on a smaller section of the reach (Figures 4, 6, 7) or for two specific discharge levels (Figures 3, 5), we think it is important to better clarify that not the full time period and reach were included in this study. We therefor added to Line 75 (Methods section): 'The full reach was studied for two contrasting discharge levels. A smaller reach (2.7 km) was studied over the full time period.'

In the abstract, we changed 'The study is based on analysis of a biweekly multibeam echosounding dataset that spans a period of approximately three years and 34 kilometers in the Waal River, a lowland sand-bedded river' to 'The study is based on analysis of a large biweekly multibeam echosounding dataset from the River Waal, a lowland sand-bedded river.

The observation that secondary bedforms become the dominant bedform scale (Figure 6) during a discharge peak cannot be extrapolated, but is only observed in a small section of the river, during a specific discharge peak. We make this more clear by adding 'locally' at several locations in the manuscript:

Line 212 (Results section): 'Figure 6 shows that*, locally,* in the southern section of the river (n=50m), at peak discharge, primary dunes fully disappear and the secondary dune scale becomes dominant.'

Line 331 (Discussion section): 'In a more extreme case, we observe that *locally*, during peak discharge, primary dunes almost fully disappear, and the superimposed dunes become the dominant bedform scale in the southern section of the river (Figure 6).'

Line 394 (Conclusions section): '*Locally*, during peak discharge, the secondary dunes are even observed to become the dominant dune scale in the southern half of the river. Once water levels drop, primary dunes reestablish.'

2) In their method the authors briefly describe criteria for identifying dunes (L151-155). However, the exclusion criteria might have a significant impact on the results of for example the coverage, dune characteristics, etc. It effectively means that authors are defining a certain area as flat bed, which they do not show in their results. It would be valuable to know if these flat bed area's are connected or exist within a dune field. Do secondary dunes persist on these flat beds or not? This might also affect the methodology: how is a dune length determined near a spatial transition to flat bed. This at least warrants a discussion.

The exclusion criteria described affect the results. This is also why the exact values of the criteria are provided in the manuscript. There exists no clear definition of when a 'fluctuation' can be considered a dune or not, which makes it difficult to determine the exclusion criteria. We have chosen to adopt quite a large range for dune height and length (compare e.g. to the median dune characteristics presented in Figure 4). Non-dimensional values (lee slope angle and aspect ratio) are based on existing literature.

The dune length is determined as the horizontal distance between an up- and downstream trough. The troughs are determined as local minima between two zero-crossings. Near a flat bed, it might occur that the local minimum differs from what we would visually determine as the dune trough. We do not expect this significantly affects the median values and IQR. After dunes have been identified, the filtering does not change the calculated characteristics of individual dunes that are not excluded. Filtering does of course change the median values. The dune identification and characterization is based on the method presented by Zomer et al., (2022), who provide a more extensive description. Results presented in Figure 4 further correspond well to dune characteristics presented by Lokin et al., (2023), who studied primary dunes in the River Waal.

The flat bed areas can be connected areas where primary dunes do not occur (e.g. as presented in Figure 6). This is reflected in the cover fraction. Where the cover fraction is very low, it is likely that there is a large flat bed area. Secondary dunes can persist over these flat bed areas, as presented in Figure 6. 'Flat bed sections' can also occur within a dune field and a.o. be the result of an error in identification leading to unrealistic values for the dune characteristics.

**Line-by-line comments:**

Figure 2: I added the discharge levels to the panels.

Line 190: Readers can see this in Figure 2. Panel d for example shows how the crest orientation of primary dunes is slanted whereas the crests of secondary bedforms are transverse to the channel axis. I added '(see a.o. Figure 2, panel d)' to line 190.

Line 199: 9. The correlation is significant ($p < 0.05$). The other dune characteristics are also significantly correlated to the discharge. I added 'For all dune characteristics, the median values significantly correlate with the discharge ($p < 0.05$).' to the caption of Figure 4.

Line 198: See response to comment 1 above.

Figure 4: Panel a shows the fraction of the river bed covered by primary dunes. Indeed, at certain timesteps, the cover is low. Here primary dunes are not identified/filtered out during processing because of the filtering criteria. In sections where not primary dunes are present, river bed is considered flat or only the secondary bedform scale is present (see e.g. Figure 6). Panel b shows the fraction of the river bed covered by secondary dunes (which are often superimposed on primary dunes, but not exclusively). For clarification, I changed the figure caption to:

'Primary and secondary dune characteristics for different discharge values. For each time step, the median of each dune characteristic was determined  for the stretch between kilometers 34.3-36. The first column shows the primary dunes characteristics, the secondary column the secondary dune characteristics. Panel a shows the fraction of the river bed covered by primary dunes (cover$_p$). If no primary dunes are present, the river bed is flat or only the secondary scale is present. Panel b shows the fraction of the river bed covered by secondary dunes (cover$_s$). Secondary bedforms are often superimposed on primary dunes, but not exclusively. Other presented characteristics include dune height (H), dune length (L), the aspect ratio ($\psi$) and the lee slope angle ($\alpha$). For all dune characteristics, the median values significantly correlate with the discharge ($p < 0.05$).'

Line 218: This line refers to Figure 7, panel e. The sentence is a bit confusion as is however, I changed '… and dunes become longer' to '… and dunes are longer compared to the northern and central section'. This better represents the data presented in Figure 7 (and Figure 5).

Figure 5: I added a legend to explain the colors. Indeed, the subplot labels do not match the caption. I changed the caption to:

'Spatial distribution of dune characteristics for two discharge level. (a) Boxplots of the median grain size, grouped based on lateral location (north, centre, south) in the river. The boxplots indicate the median, IQR and the 5$^{th}$ and 95$^{th}$ percentiles.(b-k) Lateral distribution of primary and secondary dune characteristics. Characteristics include fraction of the river bed covered by primary dunes (cover$_p$) and secondary dunes (cover$_s$), dune height (H), length (L), aspect ratio ($\psi$), and lee slope angle ($\alpha$). For each BEP, the median of each characteristic was determined over the length of the study area, as well as the IQR.'

Figure 6: I added labels to the subplots. In the caption I changed the second (d) to (e) (this was an error).

Lines 245-247: The statement regarding observations in this study are valid for the range of grain sizes presented in Figure 5 (median grain sizes of individual samples range from around 0.5 mm to 4 mm, but are mostly below 3 m,). Grain sizes in section 2A in the study by Wilbers and Ten Brinke are around 3-4 mm (Wilbers and Ten Brinke, 2003: Figure 2). Grain size might explain the difference in observations. I added 'where grain sizes are larger' to lines 241-244: 'Although Wilbers and Ten Brinke (2003) previously observed superimposed dunes specifically during the falling limb of a discharge wave at an upstream location in the Dutch part of the Rhine river*, where grain sizes are larger*, in this study, secondary dunes are clearly observed over the full range of flow conditions, including peak discharge and during stable discharge conditions (Figures 3 and 4b).'

Line 255: I added the two suggested references to line 255, as well as another study by Naqshband et al., (2017).

Line 261: Yes, see e.g. Figure 5 where in the northern river section, both dune aspect ratio and lee slope angle are larger during high discharge. We added ', which is also observed in this study' to line 261: 'Van Rijn (1984c) indicates that dune steepness increases with the transport stage parameter, for values up to around $T_{VR} = 5$, which is also observed in this study.'

**References**

Naqshband, S., Hoitink, A. J. F., McElroy, B., Hurther, D., & Hulscher, S. J. (2017). A sharp view on river dune transition to upper stage plane bed. Geophysical Research Letters, 44(22), 11-437.